# An investigation on the energy absorption characteristics of a multi-cell hexagonal tube under axial crushing loads

Li Yang[◉], Mingkai Yue[◉]*[◉], Zhen Li[‡], Tong Shen[‡], on behalf of The Chongqing postdoctoral research project[¶]

School of Equipment Engineering, Shenyang Ligong University, Shenyang, Liaoning, China

◉ These authors contributed equally to this work.
‡ These authors also contributed equally to this work.
¶ Membership of the Chongqing postdoctoral research project is provided in the Acknowledgments.
* yangliliyang321@163.com

## Abstract

A multi-cell tube enhances the energy absorption considerably compared to the absorption of a single tube under the same conditions. A novel tube configuration, namely, a multi-cell hexagonal tube, was proposed in this paper. The multi-cell tubes consist of three basic elements: a 2-panel element and two 3-panel elements (I and II). Simplified super folding element theory was utilized to estimate the energy dissipation of the basic elements. Based on this estimation, a theoretical expression for the mean crushing force was developed for the proposed tubes. The relative errors between a simulation, an experiment and theoretical results were no more than 5%. The effects of the hexagonal tube size and wall thickness on the crashworthiness of the multi-cell tubes were investigated. To a certain extent, the energy absorption and peak crushing force increased as the tube size and thickness increased. The response surface method (RSM) and the multi-objective non-dominated sorting genetic algorithm (NSGA-II) were used to improve the crashworthiness of the tube, and Pareto fronts were achieved. Finally, it was concluded that the optimal solution is C = 45 mm, t1 = 3.0 mm, and t2 = 2.35 mm, and the corresponding SEA and PCF were 16.52 kJ/kg and 411.36 kN, respectively.

## 1. Introduction

To reduce the casualties and property losses caused by train collisions, controlled dissipation of the kinetic energy of the train is crucial. Energy absorption structures need to be installed on the body frame to improve the crashworthiness of railway vehicles. Due to the large energy associated with the process of a train collision and the limited space at the front of the vehicle, multi-level energy dissipation systems are usually adopted. Designing a specific energy absorber to absorb or dissipate the prescribed energy is of great importance. A shrinking–splitting tube was used as an energy absorber for railway vehicles by Tanaskovic et al. [1]. Peng et al. [2] proposed a composite structure with diaphragms, a guide rail, and honeycomb

**Data Availability Statement:** All relevant data are within the paper and supporting information files.

**Funding:** This work is supported by Program for Innovative Talents in Institutions of Higher Education of Liaoning Province (LR2019060) and

Special support of Chongqing postdoctoral research project (XM2017115).

**Competing interests:** The authors have declared that no competing interests exist.

structures for subway vehicles, and a composite structure FE model was developed and validated by experimental data. The results also indicated that as the thickness or honeycomb yield strength increases, the initial peak force and average crashing force increase. In another study [3], four tubes with diaphragms were placed in the front end of a cab structure as an energy absorber, and it was concluded that the deformation mode was more stable than a tube without diaphragms. Xie et al. [4] designed combined thin-walled tubes with aluminium honeycombing for a railway vehicle in China based on experimental and FE simulations, and the results indicated that the entire structure generated an orderly stage-by-stage deformation.

For a long time, thin-walled metal tubes have been widely studied by many scholars through test, theory, simulation and other methods. Andrews [5] studied the mechanical properties of the tube under axial compression for the first time through theoretical and experimental methods, and obtained that the tube with different geometric parameters will produce diamond, accordion and other modes of deformation. Wierzbicki [6] proposed a theoretical model named super element method, which can effectively predict the energy absorption and average impact force during impact. Abramowicz [7] put forward a theoretical model to study the crashworthiness of square tube, and obtained the deformation modes of extension and non extension with different aspect ratio. Zhang [8] introduced a scheme of adding concave and convex surfaces on the surface of square tube to enhance the energy absorption characteristics of thin-walled square tube under axial compression. Through test and simulation, the energy absorption of the scheme was greatly improved. Gao [9] controls the number of folds in the axial buckling of the square tube by adding diaphragms in thin-walled tubes, so as to greatly improve the deformation stability and energy absorption performance. All kinds of thin-walled tubes have unique energy absorption characteristics, which are widely studied in crashworthiness field. There are a variety of ways to enhance the energy absorption of tubes, including changing the shape of the cross sections of the tubes [10], introducing a pattern into the wall [11, 12], and filling foam into the tubes [13]. Sun [14] investigated the crashworthiness circular aluminum and carbon fiber reinforced plastics (CFRP) tubes when subject to quasi-static axial and oblique compression, it was found that the energy absorption characteristics have been effectively improved. Liu [15] carried out a novel thin-walled carbon fiber reinforced plastic (CFRP) square tube filled with aluminum honeycomb, then the lateral planar crushing and bending responses has been studied, then finally come to a conclusion: the design was remarkably capable to improve the mechanical characteristics of tube. Compared with the single-cell structure, the multi-cell structure has better energy absorption capacity and axial deformation stability. Therefore, more and more valuable researches were carried out [16, 17]. Alavi Nia [18] proposed several novel multi-cell structures for 3×3 square tubes with equal or unequal cells the behaviors of which were studied analytically, experimentally and numerically. It was concluded that adding the partitions at corners can significantly increase energy absorption capacity of the tubes. Due to the difference in the basic elements, equations for the mean crushing force were deduced for each section shape. Theoretical formulas for non-filled and foam-filled double-cell and triple-cell tubes (Fig 1(A)) were put forward by Chen et al. [19], and the solution to these theoretical formulas was shown to compare very well with the numerical predictions. A set of formulas for the mean crushing force of a multi-cell profile with four square or circular tubes at the corner (Fig 1(B)) was deduced in the paper [20]. Then, the advantage of the new design over conventional single- or multi-cell profiles was discussed. Theoretical formulas for the mean crushing force of square multi-cell tubes with equal and unequal cell size were developed by Chen et al. [21] and Nia et al. [22], respectively. Optimizations were performed for single-, double-, triple- and quadruple-cell sectional tubes (Fig 1(C)) under axial crushing loading in the literature. Two different design criteria, namely, maximizing the specific energy absorption (SEA) and minimizing the peak crushing

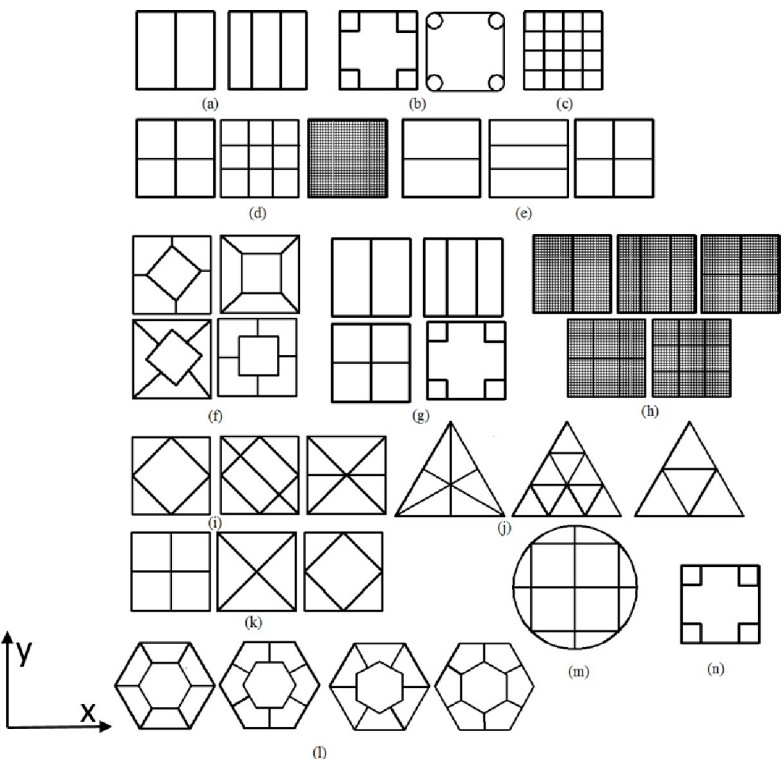

**Fig 1. Cross-sectional shape of the multi-cell tubes.**

force, were taken into account in this study [23]. Acar [24] proposed four multi-cell cross-sections by connecting the wall of the bi-tubular (Fig 1(D)) and presented an analytical mean crushing force calculation for the tubes. For these cross-sectional tubes, Jusuf et al. [25] carried out a dynamic impact test and concluded that the energy absorption efficiency can be significantly improved by introducing internal ribs to the double-walled columns. Zhang et al. [26] compared the performance of multi-cell tubes with different sections (Fig 1(E)) through quasi-static axial compression tests. The crashworthiness of six kinds of cross section structures for a functionally graded foam-filled (Fig 1(F)) was optimally designed in the paper [27]. Qiu et al. [28] derived four different hexagonal tubes with multiple cells based on the Simplified Super Folding Element (SSFE) theory through several typical constituent elements, and the results suggested that analytical formulas could be recommended in crashworthiness optimization for the sake of computational efficiency. TrongNhan Tran [29, 30] developed an equation for the mean crushing force for three types of triangular tubes with a multi-cell structure (Fig 1(G)) and three kinds of square multi-cell tubes (Fig 1(H)) under dynamic loading, and a numerical optimization of the angle element structures was carried out. Finally, the predictions coincided well with the numerical results and validated the efficiency of the numerical optimization design method. In another study [31], the energy absorption of three different configurations (Fig 1(I)) of multi-cell square tubes under oblique impact loads was investigated, and theoretical predictions of the mean crushing force were proposed. Qiu et al. [32] studied an enhanced binary particle swarm optimization of multi-cell square tubes by introducing the mass constraint factor to guide the movement of particles, which could improve the success rate of obtaining the global optimum, it was found that the optimum designs has better performance. Qiu et al. [33] proposed four different cross-sectional hexagonal multi-cell tubes (Fig 1(J)), and a set of theoretical predictions for the four tubes was derived. The authors also studied the

performance of the four tubes under oblique loading [34]. It was found that for the same cell number, the number of corners plays a significant role in enhancing the energy absorption. Zhang et al. [35] introduced a connecting flange into bi-tubal hexagonal tubes (Fig 1(K)), and an optimal design was also carried out. Stefan Tabacu [36] investigated the axial crushing behaviours of circular structures with a rectangular insert (Fig 1(L)) using both analytical and numerical methods. Fang et al. [37] introduced a wall-graded thickness to the corners of multi-cell tubes (Fig 1(M)), and the specific energy absorption of the FGT structure increased by 19.51% compared to the absorption of tubes with uniform thickness. Beik et al. [38] investigated the energy absorption of tapering S-rails with an internal diagonal reinforcement by combining reinforcing and tapering techniques. The S-rails showed a noticeable improvement in the energy absorption performance. Wu et al. [39] studied the energy absorption characteristics of five-cell tubes by experiments and numerical simulations, and the influence of the cross-sectional shape was studied. Mahmoodi [40] derived a formula for the mean crushing load of a tapered multi-cell tube and discussed the effect of the number of cells. The results revealed that an increase in the taper angle, wall thickness and number of cells in the cross-section would enhance the crashworthiness of the structure. Hou et al. [41] optimized the crushing resistance of a simple multi-cell hexagonal tube (Fig 1(N)) and found that the side-connected configuration outperforms the vertex-connected configuration. Pang [42] proposed a novel multi-cell column with axially-varying thickness (AVT), then the crushing behaviour was performed experimentally, and it was concluded that the thickness gradient of AVT multi-cell columns could effectively reduce the initial peak crushing force and increase the energy absorption.

This paper aims to investigate the energy absorption of a new multi-cell tube design which was designed to improve the crashworthiness of railway vehicles. The train buffer is always subjected to the loading in the direction of moving under the constraint of railway. According to statistics, almost all train collisions involve frontal crash. The proposed structure is a five-cell tube, as depicted in Fig 2. The structure consists of four hexagonal tubes and four connecting ribs. The length of the tube is H = 600 mm, and its cross-section dimensions are L×L = 300 mm×300 mm. According to the requirements of standards for train crashworthiness (BS EN 15227–2008) [43], the structure shall meet the frontal impact condition with the speed of 10m/s. So, this paper considered the crashworthiness under frontal impact condition to meeting the requirements of standard. The study of the structure is divided into three basic sections. First, the bending energy and membrane energy of the elements are derived, and a formula for the mean crushing force is obtained in section 2. A numerical model of the tubes is set up, and a

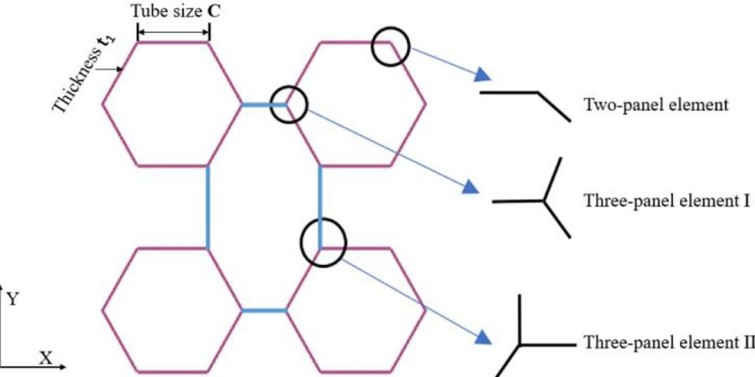

**Fig 2. Cross-sectional shape of the five-cell tube and typical angle element.**

validation of the simulation result is carried out in section 3. The effect of the wall thickness and hexagonal tube size on the crashworthiness is investigated, and the enhancement coefficients of tubes with different structural parameters are determined. Finally, the response surface method (RSM) and the multi-objective non-dominated sorting genetic algorithm (NSGA-II) is presented to achieve the optimal design under the crashworthiness criterion.

## 2. Theory

### 2.1. Theoretical model of the mean crushing force

In the initial phase of designing an energy absorber, a theoretical predication for the thin-walled structures will be of assistance to engineers. As an effective method, simplified super folding element (SSFE) theory [44, 45] is widely adopted to obtain a theoretical formula for the mean crushing load. In this paper, the SSFE method is also adopted to infer the mean crushing force of the proposed structure. A rigid-perfect plastic material is considered to describe the characteristics of the material. Based on the principle of energy conservation, the external energy dissipation is equal to the bending and membrane energy dissipation in the process of forming a single fold. That is,

$$2P_m H = \frac{1}{\delta}(E_b + E_m) \tag{1}$$

where $Pm$, $H$, $E_b$ and $E_m$ denote the mean crushing force, length of the fold, bending energy, and strain energy, respectively, and $\eta$ is the effective crushing distance coefficient. In reality, the length of the fold is smaller than 2H, because the panel is not completely flattened [46, 47]. In the analysis of this paper, the value of $\eta$ is set to 0.75, since the value was previously found between 0.7 and 0.8 [46].

**2.1.1. Bending dissipated energy.** In SSFE theory, the value of $Eb$ for each fold can be determined by summing up the energy dissipation at the three bending hinge lines [31]:

$$E_b = \sum_{i=1}^{3} M_0 \theta_i c = 2\pi M_0 L_C \tag{2}$$

where $c$ is the sectional width, $\theta$ denotes the angle of the fold, and $Lc$ is the sum of the side lengths. $M0$ is the plastic moment, which can be calculated as follows:

$$M_0 = \frac{1}{4}\sigma_0 t^2 \tag{3}$$

where $\sigma_0$ is the flow stress of the structural material and t is the wall thickness. The calculation method for the flow stress $\sigma_0$ is as follows:

$$\sigma_0 = \sqrt{\frac{\sigma_y \sigma_u}{(1+n)}} \tag{4}$$

where $\sigma_y$ is the yield strength, $\sigma_u$ is the ultimate strength, and $n$ is the strain hardening exponent for a strain-hardening material.

**2.1.2. Membrane deformation dissipated energy.** Theoretical models for the basic elements, such as the 2-panel, 3-panel and 4-panel elements, were established by Zhang et al. [48–51]. In this paper, the proposed multi-cell tube is divided into three different elements: a 2-panel element and two 3-panel elements (I and II), as shown in Fig 2. Fig 3(A) shows that the collapse pattern of the 2-panel element is an asymmetric mode. When the mode is asymmetric, the membrane energy $E_m$ of each panel is derived by integrating the extensional and

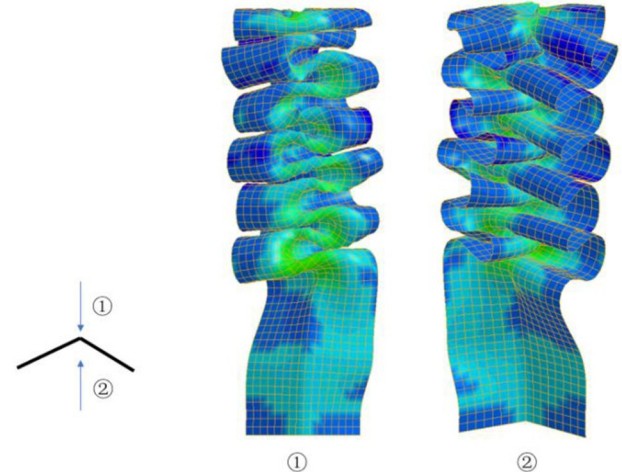

(a) The stress contours collapse of the 2-panel element

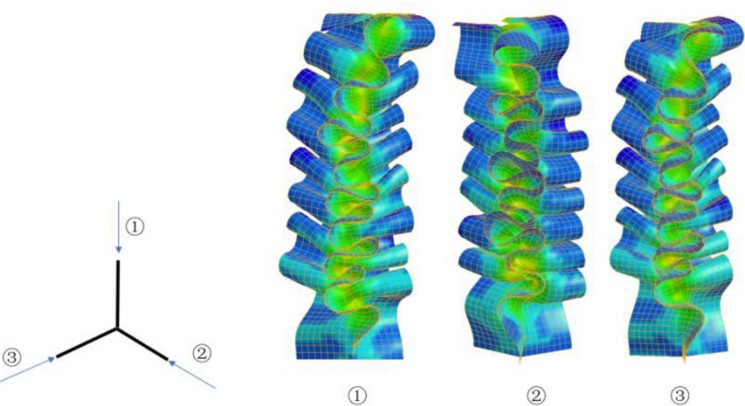

(b) The stress contours collapse of the 3-panel element I

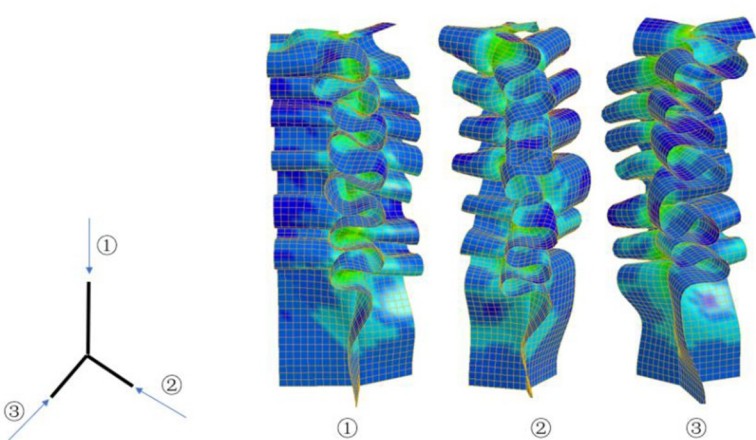

(c) The stress contours collapse of the 3-panel element II

**Fig 3. The stress contours collapse of typical angle elements.**

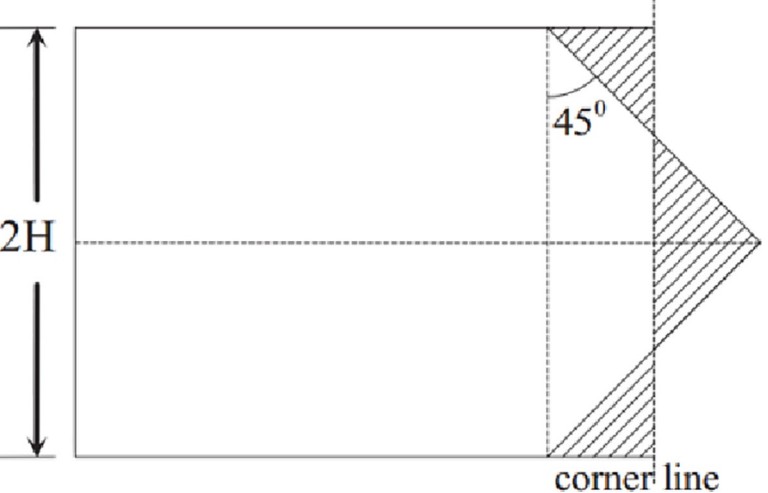

**Fig 4. Basic folding element of the asymmetric mode.**

compressional areas (shaded areas in Fig 4). Then,

$$E_m^{one-panel} = \int_s \sigma_0 t ds = \frac{1}{2}\sigma_0 t H^2 = 2M_0 \frac{H^2}{t} \tag{5}$$

For the two-panel elements, the central angle influences the membrane energy. The value of $Em$ of 120° is greater than 90° in the same situation. The central angle $\theta$ should be taken into consideration for the membrane energy of the 2-panel element. The formula for the mean crushing force of the 2-panel element is as follows:

$$E_m^{corner}(\theta) = \frac{4M_0 H^2}{t} \frac{tan(\theta/2)}{(tan(\theta/2) + 0.05/tan(\theta/2))/1.1} \tag{6}$$

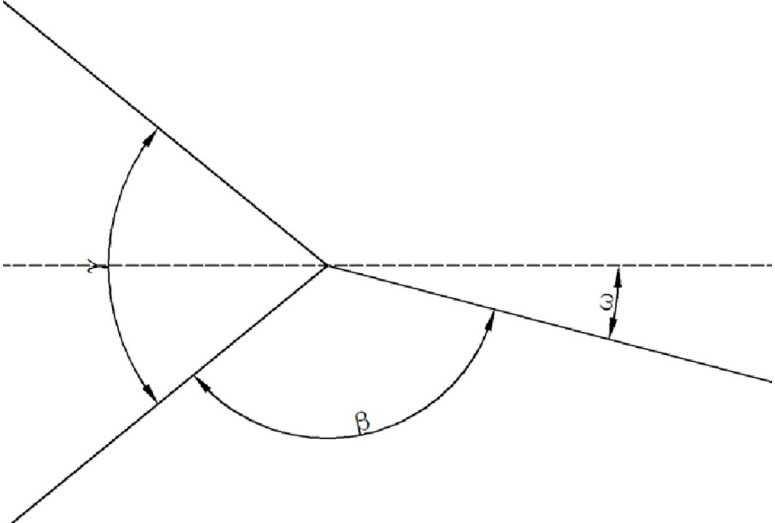

**Fig 5. Definition of the deviation angle ω for three-panel angle element II.**

3-panel element I has a symmetric plane and can be considered to consist of two parts: a 2-panel element and an additional panel. Fig 3(B) shows that the three panels are distorted in one direction, and the deformation mode of the additional panel is similar to that of the other two panels. According to the literature [48], the membrane energy of element I is equal to the sum of the two parts. Therefore, the value of $Em$ of 3-panel element I in the formation of a fold is

$$E_m^{3-I}(\alpha) = \frac{4M_0H^2}{t}\left(\frac{tan(\alpha)}{(tan(\alpha) + 0.05/tan(\alpha))/1.1} + 2tan(\alpha/2)\right) \tag{7}$$

Three-panel element II is different from three-panel element I without the symmetric plane. According to the size of the angle, the three angles can be arranged as $\varepsilon \leq \beta \leq \gamma$, as shown in Fig 5. The deviation angle $\omega$ is defined as the angle between one of the panels and the middle plane of the other two panels [48]. Fig 3(C) shows that the collapse mode of element II is similar to that of element I. The membrane energy of 3-panel element II is obtained as [48]:

$$\omega = 180° - \left(\frac{\gamma}{2} + \beta\right) \tag{8}$$

$$
\begin{aligned}
E_m^{3-II} &= E_m^A + E_m^B + E_m^C + E_m^{thickning} \\
&= 0 + \frac{8M_0H^2}{t}tan\left(\frac{\gamma}{4} + \frac{\omega}{2}\right) + \frac{4M_0H^2}{t}sin\left(\frac{\gamma}{2} + \omega\right) + \frac{4M_0H^2}{t}sin\left(\frac{\gamma}{2} - \omega\right) + \frac{8M_0H^2}{t}tan\left(\frac{\gamma}{4} - \frac{\omega}{2}\right) + \frac{12M_0H^2}{t}sin\gamma
\end{aligned} \tag{9}
$$

The basic elements in the proposed multi-cell tubes are 2-panel elements ($\theta = 120°$), 3-panel element I ($\alpha = 60°$), and 3-panel element II ($\varepsilon = 90°$, $\beta = 120°$, $\gamma = 150°$). Therefore, substituting the specific angles into Eqs (6)–(9), the values of $Em$ for three elements are expressed as

$$E_m^{corner} = \frac{4.328M_0H^2}{t} \tag{10}$$

$$E_m^{3-I} = \frac{18.1843M_0H^2}{t} \tag{11}$$

$$E_m^{3-II} = \frac{24.2265M_0H^2}{t} \tag{12}$$

**2.1.3. The formulas for the mean crushing force.** The cross-section of the tube presented in the paper is formed by a combination of sixteen 2-panel elements, four elements of 3-panel element I and four elements of 3-panel element II. To obtain a formula for the mean crushing force, Eqs (2), (10), (11) and (12) are substituted into Eq (1) to obtain the theoretical expression:

$$P_m \times 4H \times \eta = 4\pi M_0 L_c + 380.8764\frac{M_0H^2}{t} \tag{13}$$

Under the stationary condition of the mean crushing force $\frac{\partial P_m}{\partial H} = 0$, the half-wavelength can be obtained as

$$H = \sqrt{0.03299L_c t} \tag{14}$$

To substitute the term $H$ in Eq (14) into Eq (13), the mean crushing force of the proposed tube under quasi-static loading is presented as:

$$P_m = \frac{8.6478}{\eta} \sigma_0 t^{1.5} L_c^{0.5} \qquad (15)$$

## 2.2. Crashworthiness criteria

To conveniently assess the performance of the energy absorbers, it is essential to predefine crashworthiness indices. Four indices are employed in this study, including the energy absorption (EA), mean crushing force (Pm), peak crushing force (PCF), and specific energy absorption (SEA). The EA denotes the total absorbed energy of the columns in the impact process and can be expressed as:

$$EA(d) = \int_0^d F(x)dx \qquad (16)$$

where $F(x)$ is the axial crushing force and $d$ is the effective crushing displacement, which is taken as 80 mm in this study. Meanwhile, $P_m$ for a given crushing displacement $d$ can be calculated as:

$$F_m = \frac{EA(d)}{d} \qquad (17)$$

The SEA is defined as the ratio of the energy absorption to the total mass of the columns and is an important index for evaluating the energy absorption of an energy absorber. Greater values are desirable.

$$SEA = \frac{EA(d)}{m} \qquad (18)$$

where $m$ is the total mass of the column. It is clear that a higher SEA indicates a better energy absorption capacity.

## 3. Numerical investigation of the multi-cell hexagon tube

### 3.1. Finite element modelling

Finite element models are conducted to study the deformation process in this section. Fig 6 shows a schematic of the loading arrangement. The tube is impacted by the rigid wall, which has a constant velocity. The bottom of the tube is constrained in all degrees of freedom. Automatic single-surface contact is employed to consider the contact of the tube itself. The static and dynamic friction coefficients are set to 0.2 and 0.15, respectively. In order to reduce the initial peak force and trigger a stable deformation mode, indentation triggers are introduced alternately in the side panels of the tubes.

The alloy AA6060-T4 has been widely used for energy dissipation applications [7, 16]. The thin-walled tubes are assumed to be made of AA6060-T4 aluminium alloy. Model #24 (Mat_-Piecewise_Linear_Plasticity) is adopted to describe the mechanical properties. The mechanical properties of the material are as follows: the density is $\rho$ = 2700 kg/m3, Young's modulus is E = 68.2 MPa, the initial yield stress is $\sigma$y = 80 MPa, the ultimate stress is $\sigma$u = 173 MPa, Poisson's ratio $\upsilon$ = 0.3, and the power law exponent is n = 0.23 [16]. The effect of the strain rate is ignored in the numerical simulation because aluminium is insensitive to this effect [7]. The tube is modelled with a Belytchko-Tsay 4-node shell element, with five Gauss integration

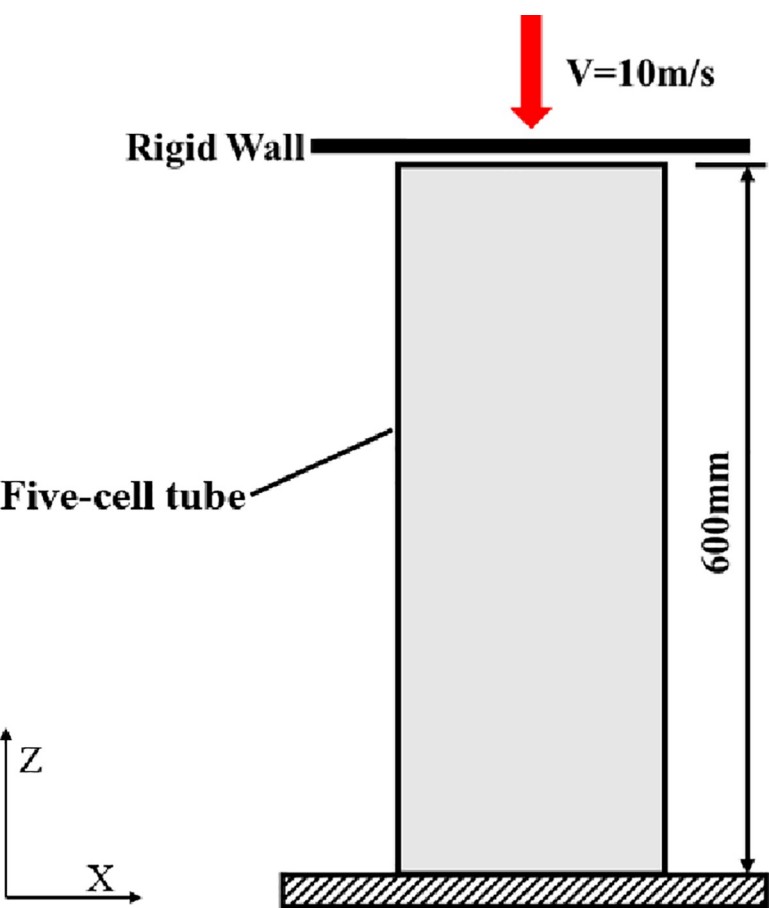

**Fig 6. Schematic of the loading arrangement.**

points through the thickness. To avoid the influence of the mesh size, the authors conduct a mesh convergence analysis and then conduct a comparative analysis of the computed results. The size of the hexagonal is 60 mm, and the thickness of the wall is 3.0 mm. Five different mesh sizes are chosen, and the results are listed in Table 1. Table 1 shows that the EA and PCF of 3.5 mm and 4.0 mm are approximately the same. Taking the computing efficiency and precision into consideration, a 4 mm mesh size is chosen. Meanwhile, the hourglass energies that occur with different densities during the computation process are all less than 2% of the total internal energy. The finite element model is shown in Fig 7. According to the literature [3], the introduction of a curved surface provides more desirable results, and the tubes are all deformed in the symmetric crushing mode. Fig 7(B) shows a view of the trigger used in the study. The remaining sides are curved with a chord 3 mm in depth.

**Table 1. Mesh convergence study results.**

|          | 3.5 mm  | 4.0 mm  | 4.5 mm  | 5.0 mm  | 5.5 mm  | 6.0 mm  |
|----------|---------|---------|---------|---------|---------|---------|
| PCF (kN) | 130.999 | 131.018 | 131.319 | 133.026 | 134.759 | 135.615 |
| EA (kJ)  | 485.804 | 487.014 | 491.263 | 493.681 | 496.264 | 498.323 |

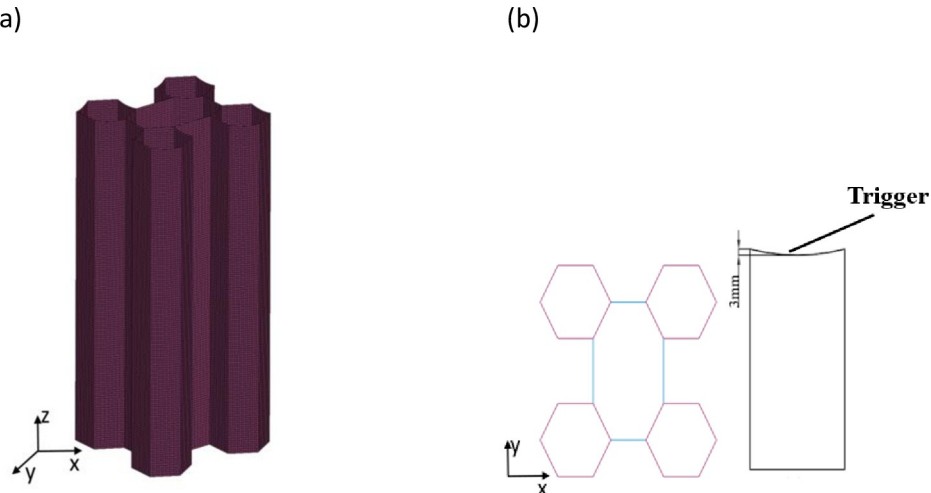

(a)												(b)

**Fig 7.** (a) Finite element model of the five multi-cell tube; (b) The trigger used in the study.

## 3.2. Validation of the FE model

It can be easily found that the multi-cell hexagonal tube will become four simple hexagonal tube when the tubes do not have a connecting wall, so we first compare the results with the common hexagonal tube to validate the simulation results. For the hexagonal tube, Zhang Xiong and Zhang Hui [26] conducted experimental studies to validate the theoretical analyses. The width, height and wall thickness of the tube are 36 mm, 120 mm and 1.2 mm, respectively. The impact speed is 1 m/s. The tube is modelled with isotropic viscoplastic aluminium. A comparison of the crushing force and deformation modes from the simulation and experimental results are plotted in Fig 8(A). The crushing force of the FE results follows the same trend as that of the experimental data. The crushing force in the FE simulation exhibits a lower value. The difference may result from the difference in the trigger sizes [27]. The stiffness of the tube is influenced by the introduced trigger. The final deformation mode for simulation model can be shown in Fig 8(B). When compared to reference [26], it can be concluded that the two deformation modes are basically the same. Table 2 shows a comparison between the numerical simulations and experimental tests. Notably, the numerical result agrees well with the experimental result.

According to previous work, the energy absorption of the structures under dynamic loading is higher than that in quasi-static situation due to inertial effects. The formula for Pm requires the introduction of a dynamic enhancing coefficient (EC) and is shown in Eq (19):

$$P_m = EC\frac{8.6478}{\eta}\sigma_0 t^{1.5}L_C^{0.5} \tag{19}$$

The EC in the research of Langseth and Hopperstad [50] was in the range of 1.3–1.6. The coefficient proposed by Na Qiu et al. [33] was 1.1. For a multi-cell square structure and circular structures with a rectangular multi-cell insert, Zhang et al. [26] and Stefan Tabacu [36] proposed a value of 1.3 for the EC. In the literature [24], for three different multi-cell triangular tubes, the coefficients were 1.41, 1.3 and 1.45. The value of the EC for the proposed multi-cell tubes is set to 1.2. Simultaneously, a comparative analysis between the theoretical expressions and the numerical results is carried out. The differences are shown in Table 3. The range of the difference is less than 5%. It can be concluded that the theoretical data agree with the computed results within the permissible error.

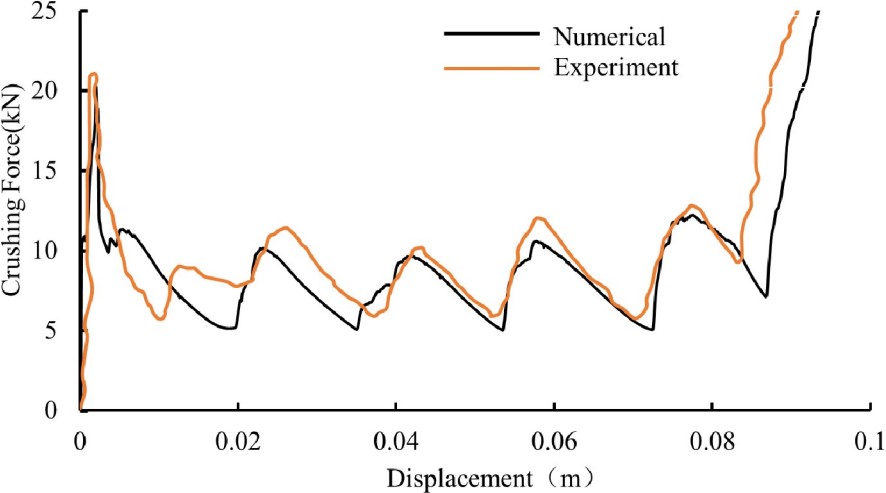

**(a) load–displacement curves**

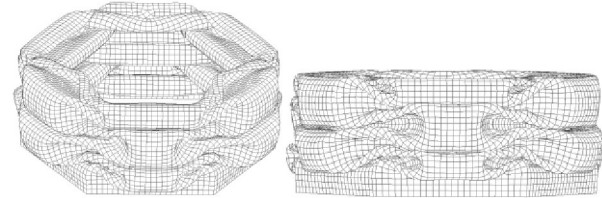

**(b) Deformation mode for model validation**

**Fig 8.** (a) Comparison between numerical and experimental. (b) Deformation mode for model validation [26].

## 4. Parametric study

The deformation modes determine the energy absorption ability of the structures, while the geometric parameters determine the deformation pattern. To obtain a proper energy absorber, considering the effect of different geometric parameters on the crashworthiness is necessary. In this part, the effects of the geometrical parameters are researched under an impact loading of 10 m/s. The parameters include the size of the hexagonal tube denoted by C, the wall thickness of the tube denoted by t1, and the wall thickness of the connecting flange denoted by t2. The size is varied from 20 mm to 65 mm. The thickness ranges from m2.0 mm to 3.0 mm.

### 4.1. Effect of the hexagonal tube size

In this section, to investigate the deformation behaviour of tubes with different hexagonal tube sizes, seven kinds of models are established. The thicknesses of the hexagonal tube and the connecting rib are the same (t1 = t2 = 2.0 mm). The deformation modes of tubes with different sizes are shown in Fig 9. For hexagonal tube sizes of 20 mm and 30 mm, the tubes develop a mixed deformation mode (local buckling and progressive buckling), while the connecting

**Table 2. Comparison between the FE simulation and experimental results [26].**

| Specimen | Experiment | | Numerical | | |
|---|---|---|---|---|---|
| | EA (J) | Pm (kN) | EA (J) | Pm (kN) | Error (%) |
| hexagonal | 759 | 9.04 | 741.4 | 8.62 | 4.6% |

**Table 3. Differences in the FE results and theoretical predictions.**

| No. | C (mm) | $t_1$ (mm) | $t_2$ (mm) | Numerical results | Theoretical results | Error (%) |
|-----|--------|-----------|-----------|-------------------|---------------------|-----------|
| 1 | 45 | 3.0 | 3.0 | 149.580 | 148.03 | 1.05 |
| 2 | 50 | 3.5 | 3.5 | 150.967 | 150.08 | 0.59 |
| 3 | 55 | 3.0 | 3.0 | 149.996 | 152.15 | 1.42 |
| 4 | 60 | 3.0 | 3.0 | 150.511 | 154.16 | 2.37 |
| 5 | 65 | 3.0 | 3.0 | 150.132 | 156.16 | 3.86 |

flange walls undergo progressive collapse. Notably, due to the small ratio of the tube size (C) to the tube length (L), the deformation is not ideal. As can be seen, for the sizes of 45 mm, 50 mm, 55 mm, 60 mm and 65 mm, a stable and progressive in-extensional deformation mode occurs. More folding lobes appear in the short connecting flanges. With an increase in the hexagonal tube size, the material of the corner position increases. Then, the fold radius will be affected due to increase in the axial stiffness of the tube, so the structural deformation is more stable. Fig 10(A) and 10(B) shows the effect of the tube size on the absorbed energy and specific energy absorption. With increasing tube size, the energy absorption first increases and then remains approximately constant. The enhancement of the absorbed energy is attributed to the progressive folding generated in the hexagonal tube regions. However, the SEA first increases with an increase in the EA and then decreases due to the increase in the mass of the tubes. The peak SEA is 12.6 kJ/kg,, which occurs for a critical corner-cell size of 40 mm, or 13.3% of the column width. For sizes larger than the critical value, the SEA decreases from the maximum due to the increase in the mass of the tubes. Fig 10(C) shows the trend of the peak crushing force for different tube sizes. The initial peak load first increases and then remains unchanged when the value of C increases.

## 4.2. Effect of the thickness on the tube

In a practical engineering application, the energy absorption is greatly affected by the wall thickness, so it is necessary to investigate the effect of the thickness on the energy absorption of the tube. Fig 11 illustrates the numerical results for nine specimens with different corner-cell wall thicknesses and connecting wall thicknesses of 2.0 mm, 2.5 mm and 3.0 mm, with the same corner-cell size of 45 mm. It is observed that the SEA and PCF increase monotonously with the wall thickness of the hexagonal tube in all circumstances. With a change in the corner-cell wall thickness (t1) from 2.0 mm to 3.0 mm and a connecting wall thickness of t2 = 2.0 mm, the energy absorption and the SEA increase from 62.8 kJ and 12.1 kJ/kg to 114.4 kJ and 16.4 kJ/kg, which are increases of 82.2% and 35.5%, respectively. Meanwhile, with the same change in the connecting wall thickness, the energy absorption and the SEA increase from 62.8 kJ and 12.1 kJ/kg to 74.7 kJ and 12.3 kJ/kg, which are increases of 18.9% and 1.6%, respectively. It can be concluded that the parameter t1 has the greatest effect on the energy absorption. From Fig 11(B), when t2 = 2.0 mm and t1 changes from 2.0 mm to 3.0 mm, the peak crushing force increases from 260.8 kN to 390.9 kN, which is an increase of 49.9%; when t1 = 2.0 mm and t2 changes from 2.0 mm to 3.0 mm, the PCF increases from 260.8 kN to 340.9 kN. Therefore, the PCF is also mainly influenced by the value of t1. This is primarily due to the increased stiffness of the tube with a thicker wall [39].

## 5. Parametric study

Since the crushing behaviours of the multi-cell tube can heavily rely on the geometric parameters, a parametric study is carried out herein to explore the influence of the geometry on the

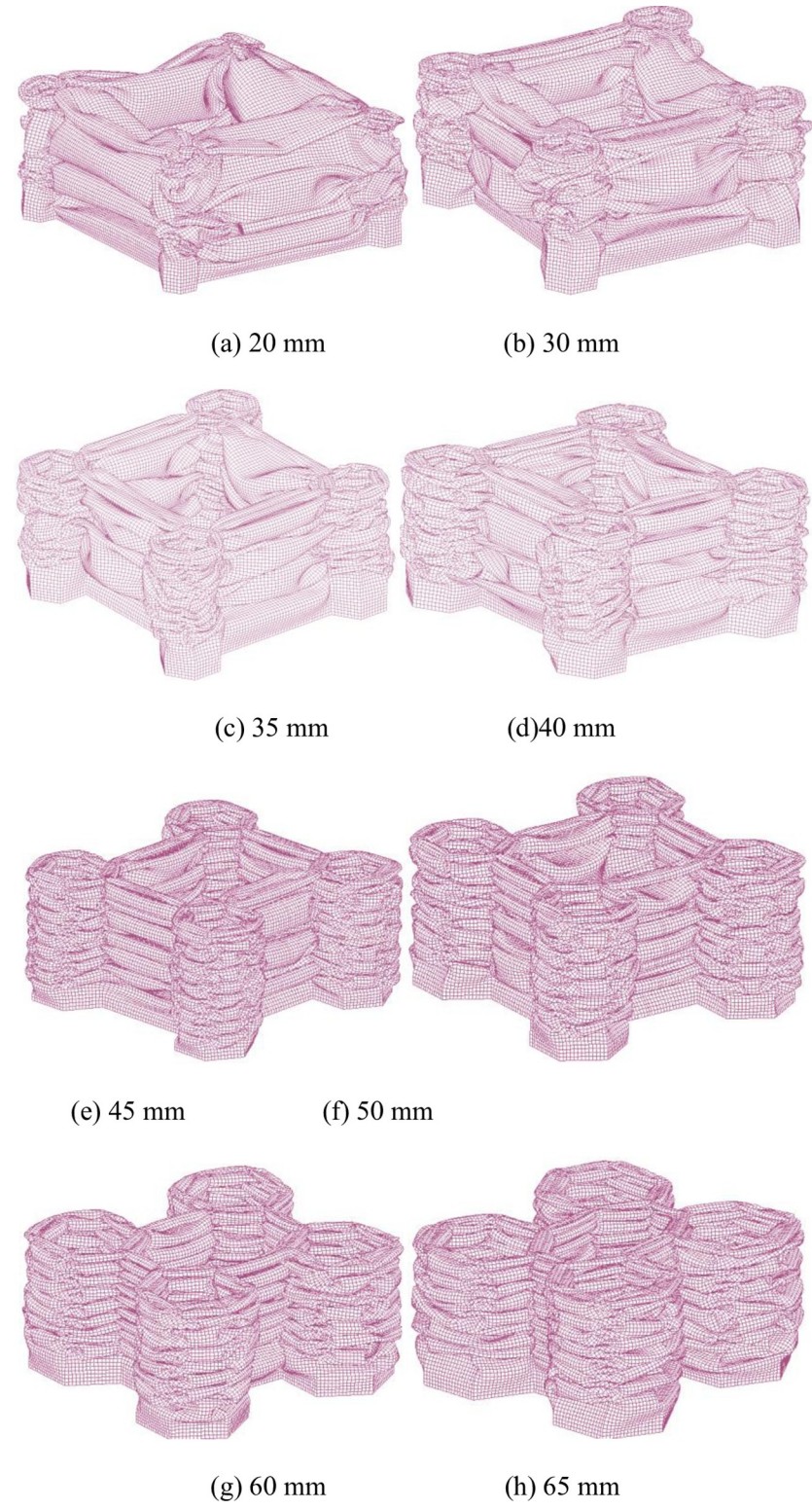

(a) 20 mm    (b) 30 mm

(c) 35 mm    (d)40 mm

(e) 45 mm    (f) 50 mm

(g) 60 mm    (h) 65 mm

**Fig 9. Deformed shapes of the tubes with different corner-cell sizes.**

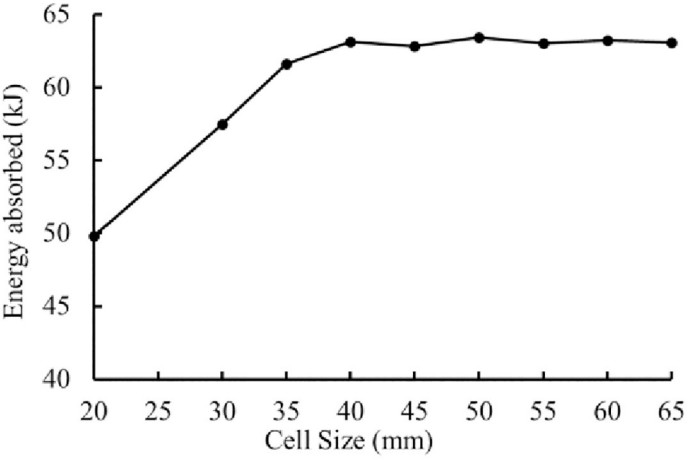

## (a) Changes of energy absorption

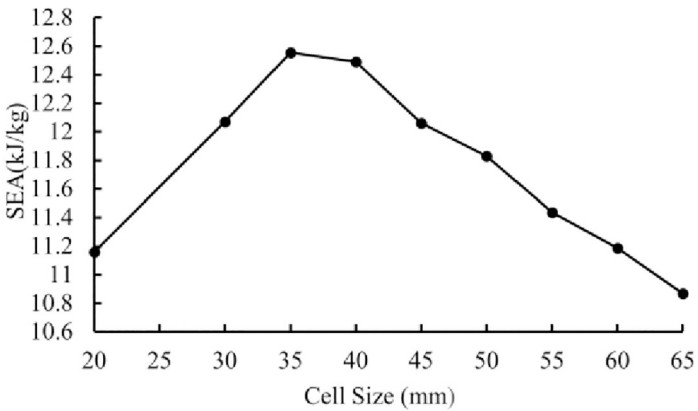

## (b) Changes of SEA

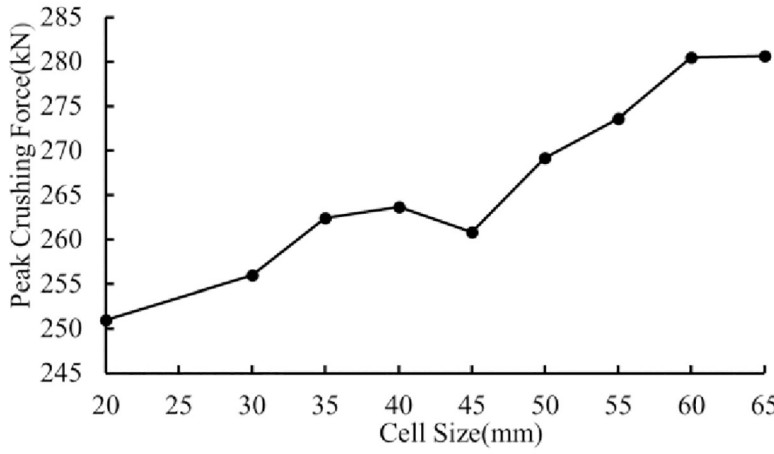

## (c) Changes of peak crushing force

**Fig 10. Effect of the corner-cell size of the energy absorption and crushing characteristics.**

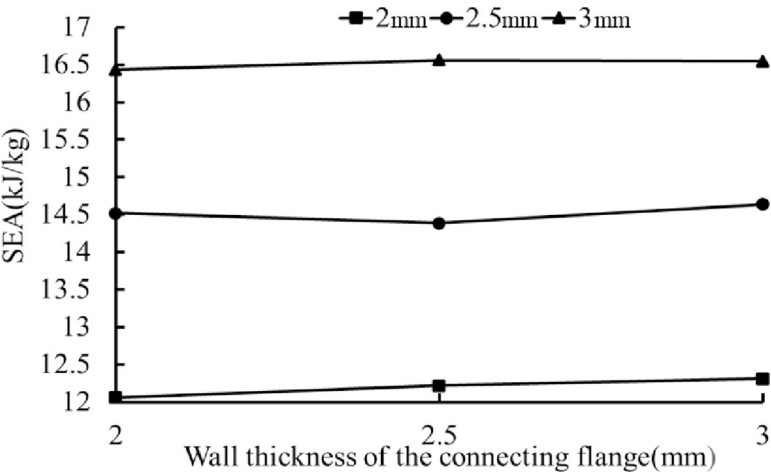

**(a) Changes of SEA**

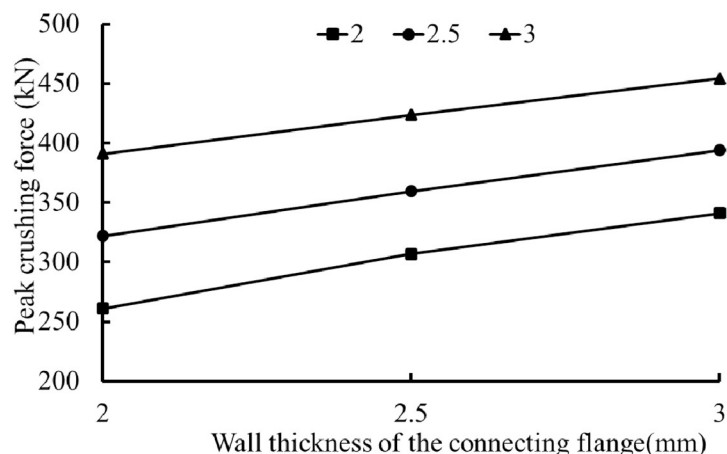

**(b) Changes of peak crushing force**

**Fig 11. Effect of the thickness on the energy absorption and crushing characteristics.**

crashworthiness under dynamic loading. The parameters include the size of the corner cell denoted by C, the wall thickness of the cell denoted by $t_1$, and the wall thickness of the connecting flange denoted by $t_2$. The size is varied from 20 mm to 65 mm. The thickness ranges from 2.0 mm to 3.0 mm.

In this section, to investigate the deformation behaviour of the tubes with different corner-cell sizes, seven kinds of models are established. An identical wall thickness (2.0 mm) is considered in each case. Examples of the collapse patterns of the tubes with different corner-cell sizes are shown in Fig 9. For corner-cell sizes of 20 mm and 30 mm, the corner cells develop a mixed deformation mode (local buckling and progressive buckling), while the connecting flange walls undergo progressive collapse. Notably, due to the small ratio of the corner-cell size (C) to the tube length (L), the deformation is not ideal. For corner-cell sizes of 45 mm, 50 mm, 55 mm, 60 mm and 65 mm, these tubes all develop a stable and progressive in-extensional deformation. More folding lobes appear in the short connecting flanges. The complex folding pattern stemming from the multi-

corner configuration depicts the interaction of a large number of folding lobes in the corner cells and connecting walls and indicates that a larger portion of the structure more uniformly participates in the plastic deformation. The connecting walls act as an elastic–plastic reinforcement for supporting the cell sidewalls, making the axial stiffness more uniform.

Fig 10(A) and 10(B) show the effect of the corner-cell size on the energy absorption characteristics. With an increase in the corner-cell size, the energy absorption increases first and then remains approximately constant. It is concluded that the progressive folding generated in the corner-cell regions results in high energy absorption. However, the SEA first increases and then decreases when the corner-cell size C increases. The peak SEA is 12.6 kJ/kg, which occurs for a critical corner-cell size of 40 mm, or 13.3% of the column width. For sizes larger than the critical value, the SEA decreases from the maximum due to the increase in the mass of the tubes. Fig 10(C) shows the trend of the peak crushing force for different corner-cell sizes. The initial peak load first increases and then remains unchanged when the corner-cell size C increases.

In practical engineering applications, the energy absorption is greatly affected by the wall thickness, so it is necessary to investigate the effect of the thickness on the energy absorption of the tube. Fig 11 illustrates the numerical results for nine specimens with different corner-cell wall thicknesses and connecting wall thicknesses of 2.0 mm, 2.5 mm and 3.0 mm, with the same corner-cell size of 45 mm. The wall thickness has a noteworthy effect on the structural crashworthiness. The SEA and PCF increase monotonously with the corner-cell wall thickness in all cases. With a change in the corner-cell wall thickness ($t_1$) from 2.0 mm to 3.0 mm and a connecting wall thickness of $t_2$ = 2.0 mm, the energy absorption and the SEA increase from 62.8 kJ and 12.1 kJ/kg to 114.4 kJ and 16.4 kJ/kg, which are increases of 82.2% and 35.5%,, respectively. Meanwhile, with the same change in the connecting wall thickness, the energy absorption and the SEA increase from 62.8 kJ and 12.1 kJ/kg to 74.7 kJ and 12.3 kJ/kg, which are increases of 18.9% and 1.6%, respectively. It can be concluded that the parameter $t_1$ has the greatest effect on the energy absorption.

From Fig 11(B), when $t_2$ = 2.0 mm and $t_1$ changes from 2.0 mm to 3.0 mm, the peak crushing force increases from 260.8 kN to 390.9 kN, which is an increase of 49.9%; when $t_1$ = 2.0 mm and $t_2$ changes from 2.0 mm to 3.0 mm, the peak crushing force increases from 260.8 kN to 340.9 kN. Therefore, the peak crushing force is also mainly influenced by the thickness of the corner-cell wall ($t_1$). This is primarily due to the increased stiffness of the tube with a thicker wall. Although ant increase in the wall thickness can improve the energy absorption, a tube with a thinner wall deforms more easily to form progressive folding and reduces the peak crashing force. Therefore, a reasonable wall thickness should be adopted in practice.

## 6. Multi-objective optimization for the tubes

An optimization of the structural parameters is an important research direction in the design of an energy absorber. The SEA and PCF are contradictory. An increase in the specific energy absorption always leads to an increase in the initial peak force. In this section, the size of the hexagonal tube C, the wall thickness of the tube t1, and the wall thickness of the connecting flange t2 are set as variables. The multi-objective optimization problem can be written as:

$$
\begin{cases}
\text{Min} & \{\text{PCF}\,(C, t_1, t_2), -\text{SEA}(C, t_1, t_2)\} \\
\text{s.t.} & 45.0\text{mm} \leq C \leq 65.0\text{mm} \\
& 2.0\text{mm} \leq t_1 \leq 3.0\text{mm} \\
& 2.0\text{mm} \leq t_2 \leq 3.0\text{mm}
\end{cases}
\tag{20}
$$

In this section, the response surface method (RSM) is chosen to approximate the value of the SEA and PCF. The RSM is an effective way to construct approximate relationships between the objectives and the design variable vector, and the analytical formulation is very complex. In this approach, the RSM approximation of the response function y1(x) is assumed as the following expression:

$$y_1 = \sum_{i=1}^{n} a_i \varphi_i(\mathbf{x}) \tag{21}$$

where n is the number of basic functions φj($\mathbf{x}$) and $\mathbf{x}$ is the vector of the normalized design variables. In general, polynomial functions are chosen as basic functions for its simplicity [52]. The simulation results for different combinations of variables are listed in S1 Table.

The selection of the basic functions should ensure sufficient accuracy and a fast convergence [31]. To find which polynomial functions better, linear, quadratic, cubic and quartic polynomials are tested in this study. The unknown coefficients $a_i$ (i = 1... n) are determined based on m design points $x_i$ (i = 1, 2...m, m>n) in the design domain. After obtaining the numerical results $\mathbf{y(x_i)}$ (i = 1,2,...m) of m design points, the method of least squares can be used to determine $a$ by the least squares of the deviation between the numerical results and the approximations. The total deviation of all design points is calculated as.

$$E(\boldsymbol{a}) = \sum_{j=1}^{m} [y(x_j) - y_1(x_j)]^2 \tag{22}$$

Subsequently, the coefficients $a_i$ (i = 1,..., n) can be determined by $\partial E / \partial a = 0$, which is written as

$$\boldsymbol{A} = (B^{\mathrm{T}} B)^{-1} B^{\mathrm{T}} y \tag{23}$$

where $B$ denotes the matrix consisting of the basic functions evaluated at the m design sampling points, which is

$$\boldsymbol{B} = \begin{bmatrix} \phi_1(x_1) & \cdots & \phi_n(x_1) \\ \vdots & \ddots & \vdots \\ \phi_1(x_m) & \cdots & \phi_n(x_m) \end{bmatrix} \tag{24}$$

To evaluate the degree of these metal models in the numerical results, the relative error (RE) is calculated as

$$RE = \frac{y_1(x) - y(x)}{y(x)} \tag{25}$$

where y(x) represents the numerical result. In this study, the R square value, the root mean squared error (RMSE) and the maximum absolute error (MAX) are also selected to evaluate the accuracies of the developed metal models.

$$R^2 = 1 - SSE/SST \tag{26}$$

$$RMSE = \sqrt{\frac{SSE}{M - p - 1}} \tag{27}$$

$$MAX = \max|y_1(x_i) - y(x_i)|, \quad i = 1, 2, \ldots, m \tag{28}$$

where the SSE and SST are the sum of the squared errors and the total sum of squares, respectively. m is the number of design points.

$$SSE = \sum_{i=1}^{M} [y_1(x_i) - y_{11}]^2 \tag{29}$$

$$SST = \sum_{i=1}^{M} [y_1(x_i) - y_{01}]^2 \tag{30}$$

where $y_{11}$ is the mean value of the approximations and y01 is the mean value of the numerical results. The corresponding parameters of the different polynomial functions are obtained by fitting the numerical results. The formulas are as follows:

**Linear:**

$$SEA(C, t1, t2) = 12.3064 - 1.6597C + 3.8647t1 + 0.3949t2$$
$$PCF(C, t1, t2) = 259.6913 + 26.4906C + 149.0044t1 + 51.6935t2$$

**Quadratic:**

$$SEA(C, t1, t2) = 12.1281 - 1.0336C + 4.3763t1 + 0.1142t2 -$$
$$0.3511C2 - 0.04570t1^2 + 0.09368t2^2 + 0.9279Ct1 + 0.3779Ct2$$
$$-0.0038t1t2$$

$$PCF(C, t1, t2) = 267.6563 - 8.7272C + 101.6168t1 + 71.5532t2 +$$
$$3.6244C2 + 17.4790t1^2 + 0.0508t2^2 + 63.8565Ct1 - 35.5786Ct2$$
$$-4.0393t1t2$$

**Cubic:**

$$SEA(C, t1, t2) = 12.1479 - 1.6036C + 4.3763t1 + 0.1142t2 - 1.2394C^2$$
$$-0.04570t1^2 + 0.09368t2^2\ 0.9279Ct1 + 0.3779Ct2 - 0.0038t1t2 -$$
$$1.0603C^3$$

$$PCF(C, t1, t2) = 268.6586 - 20.0044C + 101.6168t1 + 71.5533t2 +$$
$$83.8057C2 + 17.4790t1^2 - 0.05081t2^2 + 63.8565Ct1 - 35.5786\ Ct2$$
$$-4.0393\ t1t2 - 53.4541C^3$$

**Quartic:**

$$SEA(C, t1, t2) = 12.14343 - 0.8513C + 4.376273t1 + 0.114156t2 -$$
$$2.88306C^2 - 0.0457t1^2 + 0.093682t2^2 - 0.92788Ct1 + 0.37789Ct2$$
$$-0.0038t1t2 + 5.680071C^3 - 3.3702C^4$$

$$PCF(C, t1, t2) = 268.5824557 - 1.31452C + 101.6168t1 + 71.55325259t2$$
$$+14.26496079C2 + 17.47898\ t1^2 - 0.050806\ t2^2 + 63.85653Ct1 -$$
$$35.578578Ct2 - 4.039283t1t2 + 60.24743C^3 - 56.8507874C^4$$

where c1 is the normalized tube size and x1 and x2 are the normalized wall thicknesses. c1 =

**Table 4.** Accuracies of the different polynomial functions for the tubes.

| Objective | Order | RMSE | MAX | R2 | RE (%) |
|---|---|---|---|---|---|
| SEA | Linear | 0.1785 | 0.3889 | 0.9890 | [-3.1582,2.7155] |
| | Quadratic | 0.0972 | 0.2225 | 0.9967 | [-1.4038,1.4918] |
| | Cubic | 0.0930 | 0.2523 | 0.9970 | [-1.5561,1.7708] |
| | Quartic | 0.0914 | 0.2343 | 0.9971 | [-1.6969,1.6441] |
| PCF | Linear | 11.0722 | 29.5446 | 0.9719 | [-8.2438,8.6659] |
| | Quadratic | 2.5360 | 6.8165 | 0.9985 | [-2.6133,1.5158] |
| | Cubic | 2.1030 | 7.8188 | 0.9990 | [-2.9975,1.0154] |
| | Quartic | 2.0836 | 7.7427 | 0.9990 | [-2.9684,1.0307] |

(c-cmin)/ (cmaxcmin), and xi = (ti-timin)/ (timax-timin) (i = 1,2). The error estimations of the different polynomial functions are summarized in Table 4. Notably, the higher the order, the more approximate the estimation is. Therefore, the quartic polynomial functions of the SEA and PCF are used for the optimization. To obtain the optimal results for the tubes, the well-known non-domain sorting genetic algorithm II (NSGA-II) [44] is adopted herein. The details of the parameter definitions are summarized in Table 5. By using the metamodels and NSGA-II algorithm, the Pareto front of the multi-cell tubes under axial crushing loads for the problem defined in Eq (20) is obtained and plotted in Fig 12. It can be clearly found that an increase in the SEA always leads to an increase in the PCF. The ranges of the SEA and PCF are from 12.02 kJ/kg and 268.03 kN to 16.52 kJ/kg and 411.36 kN, respectively. The optimal hexagonal tube size and wall thickness of the two parts can be obtained from the Pareto front under different constraint conditions. When the values of the SEA and the PCF are 16.52 kJ/kg and 411.36 kN, the optimal solution is C = 45 mm, t1 = 3.0 mm, and t2 = 2.35 mm. Then, constructing the corresponding simulation model, the SEA and PCF are 16.499 kJ/kg and 409.24411.36 kN, respectively. The simulation results and optimization results are basically the same.

## 7. Conclusion

The profiles of the proposed multi-cell tubes were divided into basic elements consisting of 2- and 3-panel angle elements. Based on simplified super folding element theory, a theoretical expression for Pm was proposed for the tubes under quasi-static loading. Numerical simulations of the tubes under axial dynamic impact loading were carried out, and a parametric study was performed using finite element models. It was found that the hexagonal tube size (C), hexagonal tube wall thickness (t1) and connecting wall thickness (t2) have a distinct effect

**Table 5.** Parameter definitions for the NSGA-II study.

| Parameter | Value |
|---|---|
| Population size (multiple of 4) | 20 |
| Number of generations | 50 |
| Crossover probability | 0.9 |
| Crossover distribution index | 10 |
| Mutation distribution index | 20 |
| Initialization mode | Random |
| Maximum failed runs | 5 |
| Failed run penalty value | 1.0E30 |
| Failed run objective value | 1.0E30 |

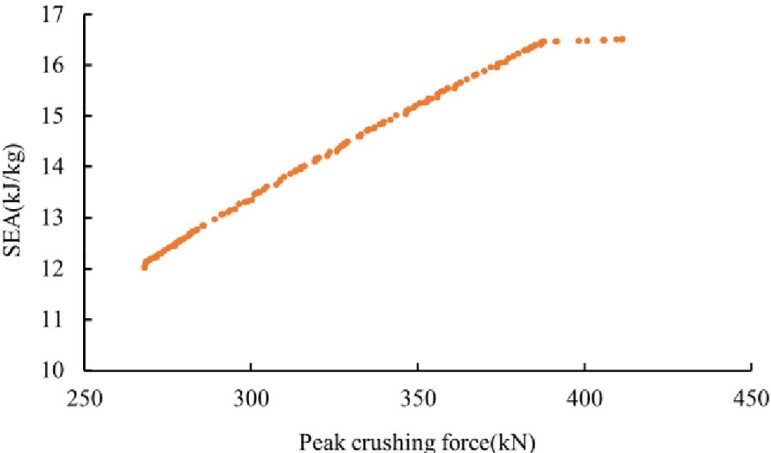

**Fig 12. The Pareto front of the multi-cell tubes under axial crushing loads.**

on the energy absorption. For a smaller ratio of the hexagonal tube size (C) to the tube length (L), the deformation is not ideal, and the hexagonal tubes develop a mixed deformation mode (local buckling and progressive buckling), while the connecting flange walls undergo progressive collapse. Stable folding deformation modes appeared for the hexagonal tubes with C = 45 mm, 50 mm, 55 mm, 60 mm and 65 mm. Through a comparison of the energy absorption of tubes with different thicknesses, it could be concluded that parameter t1 has the greatest influence on the crashworthiness. A dynamic enhancement coefficient was introduced to account for inertial effects. It could be clearly found that the value of the dynamic enhancement coefficient varies even for tubes with different geometric parameters. Multi-objective problems were formulated with respect to the hexagonal tube size (C), hexagonal tube wall thickness (t1) and connecting wall thickness (t2). A comparative study of the different polynomial functions was carried out, quartic polynomial functions were selected as the final metal model, and two RS models of the PCF and SEA were constructed. Based on the non-domain sorting genetic algorithm II (NSGA-II), optimization problems were established.

## Supporting information

**S1 Table. Crashworthiness characteristics of the multi-cell hexagonal tube.**
(PDF)

**S1 Appendix.**
(DOCX)

## Acknowledgments

This work is supported by Program for Innovative Talents in Institutions of Higher Education of Liaoning Province (LR2019060) and Special support of Chongqing postdoctoral research project (XM2017115). The financial support is gratefully acknowledged.

## Author Contributions

**Conceptualization:** Li Yang, Mingkai Yue, Zhen Li.

**Formal analysis:** Mingkai Yue.

**Funding acquisition:** Mingkai Yue, Zhen Li.

**Methodology:** Mingkai Yue.

**Project administration:** Zhen Li, Tong Shen.

**Resources:** Mingkai Yue.

**Software:** Li Yang, Mingkai Yue.

**Supervision:** Li Yang.

**Validation:** Li Yang.

**Visualization:** Li Yang, Mingkai Yue.

**Writing – original draft:** Li Yang, Mingkai Yue, Zhen Li, Tong Shen.

**Writing – review & editing:** Li Yang, Mingkai Yue, Zhen Li, Tong Shen.

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
