## [Decision Letter · Decision Letter 0]

22 Jan 2020

PONE-D-19-30455

An investigation on the energy absorption characteristics of a multi-cell hexagonal tube under axial crushing loads

PLOS ONE

Dear Prof. Mingkai Yue,

Thank you for submitting your manuscript to PLOS ONE. After careful consideration, we feel that it has merit but does not fully meet PLOS ONE’s publication criteria as it currently stands. Therefore, we invite you to submit a revised version of the manuscript that addresses the points raised during the review process.

Besides the reviewer's comments below, I suggest to present the contour plots of the strain or stress fields with the structure deformation in the revised manuscript.

We would appreciate receiving your revised manuscript by Mar 07 2020 11:59PM. To enhance the reproducibility of your results, we recommend that if applicable you deposit your laboratory protocols in protocols.io, where a protocol can be assigned its own identifier (DOI) such that it can be cited independently in the future. For instructions see: http://journals.plos.org/plosone/s/submission-guidelines#loc-laboratory-protocols

We look forward to receiving your revised manuscript.

Kind regards,

Xin Yi

Academic Editor

PLOS ONE

Journal Requirements:

2. We note that Figure 8b in your submission contain copyrighted images. All PLOS content is published under the Creative Commons Attribution License (CC BY 4.0), which means that the manuscript, images, and Supporting Information files will be freely available online, and any third party is permitted to access, download, copy, distribute, and use these materials in any way, even commercially, with proper attribution. For more information, see our copyright guidelines: http://journals.plos.org/plosone/s/licenses-and-copyright.

1.         You may seek permission from the original copyright holder of Figure(s) [#] to publish the content specifically under the CC BY 4.0 license.

Reviewers' comments:

Reviewer's Responses to Questions

**Comments to the Author**

1. Is the manuscript technically sound, and do the data support the conclusions?

Reviewer #1: Partly

2. Has the statistical analysis been performed appropriately and rigorously? 

Reviewer #1: Yes

3. Have the authors made all data underlying the findings in their manuscript fully available?

Reviewer #1: Yes

4. Is the manuscript presented in an intelligible fashion and written in standard English?

Reviewer #1: Yes

5. Review Comments to the Author

Reviewer #1: (1) The paper is very interesting and the topic of the paper falls within the scope of the journal. Overall, the paper is well organized and written.

(2) The experimental and numerical analysis are too superficial, please add some analysis on the thin-walled structures under different loading angles.

(3) Some very important pioneering works in the subject area are missing and should be well discussed in Introduction, especially some research on the thin-walled structures under crashworthiness.

(4) Please expand figure captions so figures are almost self explanatory.

6. PLOS authors have the option to publish the peer review history of their article (what does this mean?). If published, this will include your full peer review and any attached files.

Reviewer #1: No

---

## [Author Response · Author response to Decision Letter 0]

6 Apr 2020

Response to the reviewer 

Comment1: The paper is very interesting and the topic of the paper falls within the scope of the journal. Overall, the paper is well organized and written.

Response: Thank you for your constructive comments concerning our manuscript. 

Comment2: The experimental and numerical analysis are too superficial, please add some analysis on the thin-walled structures under different loading angles.

Response: Thanks for your suggestion. It is really true as reviewer suggested that the loading conditions are too superficial. The energy absorber proposed in this paper was designed to improve the crashworthiness of railway vehicles. The train buffer is always subjected to the loading in the direction of moving under the constraint of railway. According to statistics, almost all train collisions involve frontal crash. According to the requirements of standards for train crashworthiness(BS EN 15227-2008), the structure shall meet the frontal impact condition with the speed of 10m/s. So, this paper only considered the crashworthiness under frontal impact condition to meeting the requirements of standard. Oblique loading condition involves complex multi-directional instability, which will be our future work direction. We have added the following explanation in manuscript:

This paper aims to investigate the energy absorption of a new multi-cell tube design which was designed to improve the crashworthiness of railway vehicles. The train buffer is always subjected to the loading in the direction of moving under the constraint of railway. According to statistics, almost all train collisions involve frontal crash.

According to the requirements of standards for train crashworthiness(BS EN 15227-2008) [43], the structure shall meet the frontal impact condition with the speed of 10m/s. So, this paper considered the crashworthiness under frontal impact condition to meeting the requirements of standard. 

[43] Technical Committee CEN/TC 256 Railway Applications, BS EN 15227-2008 Railway Applications: Crashworthiness Requirements for Railway Vehicle Bodies, British Standard Institution, London, 2008.

Comment3: Some very important pioneering works in the subject area are missing and should be well discussed in Introduction, especially some research on the thin-walled structures under crashworthiness.

Response: Considering the Reviewer’s suggestion, we have revised the Introduction and literature review. The application of thin-walled structures in the energy absorption system has been added in the manuscript. The pioneering works have also been introduced through the existing literature. The added literatures are as follows:

[5] Andrews K R F, England G L, Ghani E . Classification of the axial collapse of cylindrical tubes under quasi-static loading[J]. International Journal of Mechanical Sciences, 1983, 25(9-10):687-696.

[6] Wlodzimierz, Abramowicz, and, et al. Dynamic axial crushing of square tubes[J]. International Journal of Impact Engineering, 1984.

[7] Abramowicz W, Jones N. Dynamic axial crushing of circular tubes[J]. International Journal of Impact Engineering, 1984, 2(3):263-281.

[8] X. Zhang, G. Cheng, A comparative study of energy absorption characteristics of foam-filled and multi-cell square columns, Int. J. Impact Eng. 34 (11) (2007)

1739–1752.

[9] Gao G, Dong H, Tian H. Collision performance of square tubes with diaphragms[J]. Thin-Walled Structures, 2014, 80:167-177.

[14] Guangyong, S. On crashing behaviors of aluminium/CFRP tubes subjected to axial and oblique loading: An experimental study[J]. Composites Part B: Engineering, 2018.

[15] Liu Q, Xu X, Ma J. Lateral crushing and bending responses of CFRP square tube filled with aluminum honeycomb [J]. Composites, 2017, 118B(JUN.):104-115.

[18] Nia A, Kazemi M. Experimental study of ballistic resistance of sandwich targets with aluminum face-sheet and graded foam core[J]. Journal of Sandwich Structures & Materials, 2018:109963621875766.

[28] Qiu N , Gao Y , Fang J , et al. Theoretical prediction and optimization of multi-cell hexagonal tubes under axial crashing[J]. Thin-Walled Structures, 2016, 102:111-121.

[32] Qiu N , Gao Y , Fang J , et al. Topological design of multi-cell hexagonal tubes under axial and lateral loading cases using a modified particle swarm algorithm[J]. Applied Mathematical Modelling, 2018, 53(JAN.):567-583.

[42] Pang T, Zheng G, Fang J, Energy absorption mechanism of axially-varying thickness (AVT) multicell thin-walled structures under out-of-plane loading. Engineering Structures 2019;196:112-122.

Comment4: Please expand figure captions so figures are almost self explanatory.

Response: We have improved our paper according to the reviewer's comment. The Fig.1, Fig.2, Fig.3, Fig.6 Fig.7, Fig.8, Fig.10, Fig.11 and Fig.12 have been revised.

Once again, thank you very much for your comments and suggestions.

---

## [Decision Letter · Decision Letter 1]

12 May 2020

An investigation on the energy absorption characteristics of a multi-cell hexagonal tube under axial crushing loads

PONE-D-19-30455R1

Dear Dr. Yue,

We are pleased to inform you that your manuscript has been judged scientifically suitable for publication and will be formally accepted for publication once it complies with all outstanding technical requirements.

With kind regards,

Xin Yi

Academic Editor

PLOS ONE

Additional Editor Comments (optional):

Reviewers' comments:

Reviewer's Responses to Questions

**Comments to the Author**

1. If the authors have adequately addressed your comments raised in a previous round of review and you feel that this manuscript is now acceptable for publication, you may indicate that here to bypass the “Comments to the Author” section, enter your conflict of interest statement in the “Confidential to Editor” section, and submit your "Accept" recommendation.

Reviewer #1: All comments have been addressed

2. Is the manuscript technically sound, and do the data support the conclusions?

Reviewer #1: Yes

3. Has the statistical analysis been performed appropriately and rigorously? 

Reviewer #1: Yes

4. Have the authors made all data underlying the findings in their manuscript fully available?

Reviewer #1: Yes

5. Is the manuscript presented in an intelligible fashion and written in standard English?

Reviewer #1: Yes

6. Review Comments to the Author

Reviewer #1: The authors have addressed my comments completely and hence I suggest the paper to be published in this current revised form.

7. PLOS authors have the option to publish the peer review history of their article (what does this mean?). If published, this will include your full peer review and any attached files.

Reviewer #1: No

---

## [Editor Report · Acceptance letter]

28 May 2020

PONE-D-19-30455R1 

An investigation on the energy absorption characteristics of a multi-cell hexagonal tube under axial crushing loads 

Dear Dr. Yue:

I am pleased to inform you that your manuscript has been deemed suitable for publication in PLOS ONE. Congratulations! Your manuscript is now with our production department. 

With kind regards,

on behalf of

Dr. Xin Yi 

Academic Editor

PLOS ONE